# Alterations in Plasma Lipid Profiles Associated with Melanoma and Therapy Resistance

**DOI:** 10.3390/ijms25031558

**Published:** 2024-01-26

**Authors:** Michele Dei Cas, Chiara Maura Ciniselli, Elisabetta Vergani, Emilio Ciusani, Mariachiara Aloisi, Valeria Duroni, Paolo Verderio, Riccardo Ghidoni, Rita Paroni, Paola Perego, Giovanni Luca Beretta, Laura Gatti, Monica Rodolfo

**Affiliations:** 1Clinical Biochemistry and Mass Spectrometry Laboratory, Health Sciences Department, Università degli Studi di Milano, 20122 Milan, Italy; michele.deicas@unimi.it (M.D.C.); riccardo.ghidoni@unimi.it (R.G.); rita.paroni@unimi.it (R.P.); 2Unit of Bioinformatics and Biostatistics, Department of Epidemiology and Data Science, Fondazione IRCCS Istituto Nazionale dei Tumori, 20133 Milan, Italy; chiara.ciniselli@istitutotumori.mi.it (C.M.C.); valeria.duroni@istitutotumori.mi.it (V.D.); paolo.verderio@istitutotumori.mi.it (P.V.); 3Unit of Translational Immunology, Department of Experimental Oncology, Fondazione IRCCS Istituto Nazionale dei Tumori di Milano, 20133 Milan, Italy; mariachiara.aloisi@uni-wuerzburg.de (M.A.); monica.rodolfo@istitutotumori.mi.it (M.R.); 4Department of Diagnostic and Technology, Fondazione IRCCS Istituto Neurologico Carlo Besta, 20133 Milan, Italy; emilio.ciusani@istituto-besta.it; 5Molecular Pharmacology Unit, Department of Experimental Oncology, Fondazione IRCCS Istituto Nazionale dei Tumori, 20133 Milan, Italy; paola.perego@istitutotumori.mi.it; 6Laboratory of Neurobiology and UCV, Neurology IX Unit, Fondazione IRCCS Istituto Neurologico Carlo Besta, 20133 Milan, Italy; laura.gatti@istituto-besta.it

**Keywords:** lipid metabolism, melanoma, drug resistance, target therapy, plasma lipid profiles

## Abstract

Dysfunctions of lipid metabolism are associated with tumor progression and treatment resistance of cutaneous melanoma. BRAF/MEK inhibitor resistance is linked to alterations of melanoma lipid pathways. We evaluated whether a specific lipid pattern characterizes plasma from melanoma patients and their response to therapy. Plasma samples from patients and controls were analyzed for FASN and DHCR24 levels and lipidomic profiles. FASN and DHCR24 expression resulted in association with disease condition and related to plasma cholesterol and triglycerides in patients at different disease stages (*n* = 144) as compared to controls (*n* = 115). Untargeted lipidomics in plasma (*n* = 40) from advanced disease patients and controls revealed altered levels of different lipids, including fatty acid derivatives and sphingolipids. Targeted lipidomics identified higher levels of dihydroceramides, ceramides, sphingomyelins, ganglioside GM3, sphingosine, sphingosine-1-phosphate, and dihydrosphingosine, saturated and unsaturated fatty acids. When melanoma patients were stratified based on a long/short-term clinical response to kinase inhibitors, differences in plasma levels were shown for saturated fatty acids (FA 16:0, FA18:0) and oleic acid (FA18:1). Our results associated altered levels of selected lipid species in plasma of melanoma patients with a more favorable prognosis. Although obtained in a small cohort, these results pave the way to lipidomic profiling for melanoma patient stratification.

## 1. Introduction

Cutaneous melanoma displays a rising worldwide incidence, but in the past decade, its clinical management has been revolutionized with the introduction of BRAF and MEK inhibitors (BRAFi and MEKi) and of immunotherapy with antibodies to immune checkpoint inhibitors, achieving a dramatic improvement of patient overall survival. Several new drugs and therapeutic options have been studied and approved, together with drug combination strategies and adjuvant and neoadjuvant therapy settings, showing promising safety and efficacy results compared to conventional chemotherapy [1,2].

Metastatic disease progression and drug resistance are associated with metabolic reprogramming and dysregulated lipid metabolism [3,4]. Alterations of lipid regulatory pathways, including pathways controlling cholesterol (CHOL), fatty acid (FA) synthesis [5,6,7], and sphingolipids (SL) metabolism [8,9], have been associated with disease progression and resistance to kinase inhibitors (KI), both in patients’ tumors and in cell lines model systems.

We recently reported that transcriptomes of BRAFi/MEKi-resistant melanoma are characterized by dysregulated lipid metabolism pathways involving FA and CHOL [10] and determining alterations in the lipid composition, particularly in different classes of FA, cholesteryl esters (CE) and triglycerides (TG), along with modulated expression of enzymes regulating biosynthetic nodes of the lipid metabolism [11]. The effect of tackling lipid metabolism pathways in resistant cell lines was evidenced by lipid starvation, which reduced cell growth and increased sensitivity to the BRAFi PLX4032/vemurafenib [11]. Specifically, we showed that the combination of BRAFi with inhibitors of lipogenic enzymes fatty acid synthase (FASN) and 24-dehydrocholesterol reductase (DHCR24) resulted in favorable drug interactions, potentially exploitable for the treatment of BRAFi-resistant melanoma [11,12].

Alterations in lipid profiles, which elucidate disease mechanisms and represent valuable biomarkers, are expected to have a pivotal role in patients’ risk stratification as well as in precision medicine. These features allow for the setting up of medical treatments based on individual lipid profiles, namely the ‘lipidome’. Specifically, the ‘lipidome’ is dependent on two factors: the biological properties of cells, which reflect the genomic alterations in biological systems, and the environmental impact on those biological systems. By capturing the complex, multilayered landscape of the lipidome as well as tracking changes in lipid levels and composition, lipidomics can reveal distinct molecular phenotypes that indicate the physiological and pathological state of an organism [13,14,15,16].

In this explorative study, we aimed to evaluate the involvement of regulatory lipid pathways at the plasma level in melanoma and in drug resistance. We quantitatively assessed the levels of FASN and DHCR24 enzymes in melanoma patients and healthy control subjects in relation to plasma TG and CHOL and tested plasma lipid composition by untargeted/targeted lipidomic approaches. In addition, to uncover potential alterations of lipid species that specifically characterize the signature of melanoma KI-resistance, we quantitively determined the lipid species in plasma from patients showing a long or a short response to targeted therapy.

## 2. Results

### 2.1. Plasma Levels of DHCR24 and FASN Enzymes Are Altered in Melanoma Patients Compared to Control Subjects

Our previous studies showed that transcriptomes of BRAFi/MEKi-resistant melanoma cells are characterized by dysregulated lipid metabolism pathways involving FA and CHOL, determining alterations in the lipid composition, along with modulated expression of enzymes regulating biosynthetic nodes of the lipid metabolism. FASN and DHCR24 lipogenic enzymes were shown to be involved in the modulation of BRAFi response in melanoma [11,12]. Based on these findings, we explored the plasma levels of FASN and DHCR24 as potential biomarkers in melanoma patients and tested the association between the expression levels of the two enzymes and the different clinical-demographic features of melanoma patients, including blood total CHOL, high-density lipoprotein (HDL), low-density lipoprotein (LDL) and TG. Quantitative analysis was carried out in a cohort of 144 melanoma patients at different clinical stages (stage I–IV) and 115 control subjects, which comprised individuals at surgery for benign skin lesions and healthy blood donors. Table 1 reports the main clinical-demographic characteristics of the study cohort.

By considering all the tested plasma samples, the levels of DHCR24 (ANOVA *p*-value 0.019) and FASN (ANOVA *p*-value 0.011) were statistically associated with disease condition (Figure 1A,B).

In detail, by looking at individuals undergoing surgery for skin lesions suspected of melanoma, in subjects diagnosed with melanoma by pathological analysis (PT DH I/II), the plasma levels of FASN were shown to be significantly higher compared to all the other groups (Tukey *p*-values < 0.01, Figure 1C). Conversely, an opposite trend was shown for DHCR24 levels. The only significant association after multiple comparison adjustments was shown in patients at advanced stages (PT III/IV) vs. subjects at surgery of benign skin lesions (control DH) (Tukey *p*-value 0.017, Figure 1D). By looking at the logistic regression results, the odds ratio for FASN and DHCR24, as well as for other biochemical variables in a univariate fashion, showed significance for both FASN and DHCR24 (Table 2), although when adjusting for age, neither of these markers retained the statistical significance. No correlation between FASN and DHCR24 levels was observed. In addition, by looking at the pair-wise correlations of FASN and DHCR24 with the other variables, we observed a low magnitude pattern of correlation: FASN levels were positively correlated with CHOL, LDL, and TG, whereas DHCR24 levels were negatively correlated to TG (Appendix A).

We next analyzed the potential association of FASN and DHCR24 with treatment response duration in a group of advanced metastatic patients at the baseline of medical treatments resulting in a long- (long-term responders, LR) or a short- (short-term responders, SR) term clinical response (Table 1). Higher plasma levels of FASN were observed in LR patients (Figure 1E) with a borderline significance (OR: 1.571; 95%CI: 0.976; 2.528), whereas no association was observed between DHCR24 and treatment response (Figure 1F) and with the other considered plasma variables (Appendix A). The same trend was observed in patients treated with BRAFi/MEKi (KI, *N* = 40) and in those treated with immunotherapy with immune checkpoint inhibitors (ICI, *N* = 40) (Figure 1G). The opposite behavior was observed for DHCR24, showing higher levels in SR patients (Figure 1H), while no significant association was observed between the treatment response duration and the other plasma variables considered.

### 2.2. Lipidomic Profiling of Plasma Reveals Altered Levels of Fatty Acids Derivatives and of Sphingolipids in Melanoma Patients

To evaluate potential alterations of systemic lipid metabolism in melanoma progression, we carried out an untargeted lipidomic analysis in 20 plasma samples from advanced melanoma patients at the baseline of targeted therapy with KI and in 20 sex-matched control subjects (control DH). Among the 935 lipid species recognized, case-control analysis at a single lipid level identified four lipids associated (after false discovery rate—FDR-adjustment) with the disease condition and showing lower values in patients, including glycerolipids and glycerophospholipids TG 56:0 (or TG 16:0/18:0/22:0), DG 25:4, DG 50:3, and PC O-39:10 (Figure 2A).

At the lipid family level, case-control analysis highlighted three families with a different expression pattern between cases and control, with free FA showing higher levels in patients compared to controls and hydroxylated phytosphingolipids Cer AP (alpha-hydroxy fatty acid phytoceramide) and HexCer AP (alpha-hydroxy fatty acid phytohexosylceramide) showing lower levels in patients (Figure 2B). In LR compared to SR patients, diacylglycerol (DG 32:5) and FA derivatives N acyl glycil serine (NAGly Ser) 25:0, CAR 26:0, and vitamin A fatty acid ester (VAE) 13:0 showed significantly higher values, while sphingomyelin SM 33:2 was under-expressed (Figure 2C). None of the latter was included in the list of lipids differentially expressed between cases and controls, suggesting a possible involvement of different lipids in the two scenarios.

### 2.3. Targeted Analysis of Fatty Acid and Sphingolipid Species Identify Quantitative Differences in Patients Responsive to Targeted Therapy

To deepen the analysis, we performed a targeted lipidomic quantification focused on FA and SL profiles. What appeared striking about the complex plasma SL profile is their diffuse increased concentrations in melanoma patients compared to control subjects (Figure 3A). The more remarkable differences were appreciated in the total concentrations of dihydroceramides (DHCer, FRD-W *p*-value: 0.006), ceramides (Cer, FRD-W *p*-value: 0.004), sphingomyelins (SM, FRD-W *p*-value: 0.001), and gangliosides (GM3, FRD-W *p*-value: <0.001). When we compared samples from patients classified according to response to targeted therapy, the differences in the SL pathway were less marked and lacked statistical significance (Figure 3B).

Relatively to sphingoid bases, a sharp difference was found in the concentration of the catabolic products sphingosine (Sph, FRD-W *p*-value: 0.009) and sphingosine-1-phosphate (S1P, FRD-W *p*-value: 0.011) as well as dihydrosphingosine (dhSPh, FRD-W *p*-value: <0.001), which is an early intermediate of the de-novo SL biosynthesis (Figure 4A). By contrast, no significant differences were found between SR and LR.

For the analysis of the plasma levels of free FA, we focused only on FA arising from the direct or related activity of FASN, which drives the synthesis of palmitate. The pattern found for the SL concentrations was also observed for the FA plasma content, as higher levels are found in melanoma patients (Figure 4B). This trend reaches a borderline significance after FRD adjustments for stearic FA (FA 18:0, FRD-W *p*-value: 0.065) and for the unsaturated derivatives oleic (FA 18:1, FRD-W *p*-value: 0.065), linoleic (FA 18:2, FRD-W *p*-value: 0.058), linolenic (FA 18:3, FRD-W *p*-value: 0.025) and palmitoleic acids (FA 16:1, FRD-W *p*-value: 0.072), while no significance was instead observed for palmitic acid (FA 16:0, FRD-W *p*-value: 0.236). In patients stratified based on treatment response, a trend towards different FA plasma levels was shown for the saturated FA 16 and FA 18 (W *p*-value: 0.014) and oleic acid (FA 18:1, W *p*-value: 0.11), which were higher in the group of responsive patients although no significances after FDR adjustments were reached (Figure 4C). Of note, superimposable results were obtained when lipid families were computed according to the Principal Component Approach (PCA).

## 3. Discussion

Several lines of evidence implicate the dysregulation of lipid metabolism in melanoma progression and therapy resistance [17]. Subcutaneous adipose tissue and released adipokines were suggested to promote melanoma growth [18]. Melanoma cells were shown to be able to take up lipids from extracellular vesicles released by stromal adipocytes or from aged fibroblasts by fatty acid transport protein (FATP) and use those to survive upon targeted therapy [19,20,21]. Dysregulation of CHOL homeostasis in melanoma cells was shown to impact proliferation, migration, and invasion, and overexpression of CHOL synthesis genes was associated with poor disease prognosis and drug resistance [10,22]. High prostaglandin E2 (PGE2) synthesis and increased FA oxidation are associated with poor prognosis in melanoma patients after targeted therapy with BRAFi/MEKi [6,21,23]. The expression of genes implicated in FA metabolism was shown to be associated with the expression of melanocyte inducing transcription factor (MITF), a main regulator of the proliferative to invasive phenotype switch in melanoma cells, and to AXL receptor, a hallmark of a de-differentiated and drug-resistant phenotype [2,7].

As reported in previous studies carried out in our lab, vemurafenib-resistant melanoma cell line models displayed increased levels of key enzymes controlling lipid metabolism [10,12]. Drug-induced cell selection importantly impacts on lipid assets, as resistant cell lines revealed an altered lipid composition, characterized by a general increase in mono-unsaturated fatty acids (MUFA) counterbalanced by a decrease in SFA in FA as well as in TG and CE. In addition, a trend for the increase in FA typical of SM (C20:0, C20:1, C20:2, C22:0, C24:0, and C24.1) was also evidenced [11]. When quantitatively evaluated by SL targeted analysis, the total amount of Cer (d18:1, FA from C14 to 24) and SM (d18:1, FA from C16 to 24) as well as Sph d18:1 was higher in resistant cells compared to their sensitive counterparts and appeared modulated by drug treatment. Several studies showed that SL deregulated metabolism results in the accumulation of metabolites endowed with a pro-tumoral action, including both simple Cer derivatives, including Sph, as well as complex SL, such as sphingomyelin and GM3 [9,24].

Lipidomics is the most powerful analytical tool for studying lipids in biological samples and their biochemical involvement in human diseases. In particular, bioactive lipids are considered novel biomarkers that may contribute to innovative medical treatments, early intervention, and improved patients’ prognoses. Therefore, the identification of lipidomic signatures may be relevant for the detection of potential diagnostic, prognostic, and predictive biomarkers of cancer [25]. In melanoma, examining the concentrations of a subset of lipids may shed light on cellular processes and reveal changes in targetable pathways [26].

Recently, the profiles of plasma lipidome in cancer patients have gained attention for potential translation from basic research to clinical application [27]. In this pilot experimental study, we evaluated the levels of lipogenic enzymes and lipid profiles in plasma from melanoma patients compared to healthy controls and their potential association with clinical response to treatment. As the levels of the lipogenic enzymes FASN and DHCR24 differed in melanoma patients with respect to controls, we proceeded further to lipidomic profiling studies. To this purpose, targeted and untargeted lipidomic approaches were applied. Untargeted lipidomics indicated that three major lipid families showed quantitative differences in patients, including phytosphingolipids (CerAP and HexCerAP) and FA, with the latter showing an enrichment (Figure 2B). A targeted approach towards complex SL, sphingoid bases, and free FA (saturated and unsaturated) families was then applied to narrow the analysis of these specific lipid species regulated in the CHOL and FA biosynthetic pathways. We found higher levels of SL DHCer, Cer, SM, HexCer, LacCer, and GM3, sphingoid bases (e.g., Sph, S1P, and dhSph) and of free FA (e.g., FA 16:1, 18:0, 18:1, 18:2, 18:3) in patients compared to controls (Figure 3 and Figure 4). Concerning the FA, higher levels of FA 16:0, 18:0, and 18:1 were evidenced in LR in comparison to SR (Figure 3 and Figure 4). Notably, with respect to SR, LR patients showed a trend for higher levels of DHCer, Cer, SM, and LacCer, while the levels of GM3 revealed an opposite trend with higher values in SR. Gangliosides, including the simplest GM3, are present in melanoma cells in high amounts and have been associated with melanoma progression [28]. The higher levels of GM3 found in melanoma patients as compared to controls and higher levels evidenced in poorly responsive patients suggest a possible different expression in melanoma.

In summary, the levels of specific SL, sphingoid bases, and saturated/unsaturated free FA defined a lipid pattern distinguishing advanced melanoma patients. This specific lipid pattern, together with the plasma level of FASN and DHCR24, stratifies advanced patients in SR and LR, suggesting a potential predictive value for treatment response. Although interesting, these results were obtained in a small sample set and require further validation studies in larger cohorts.

Our results are in agreement with data reported by Garandeu [29], showing higher plasma levels of individual long-chain Cer species C22:0, C24:0, and C24:1 in three patients classified as responders compared to 3 non-responders to BRAFi, suggesting that high levels of SL may distinguish responsive patients. While no evidence for a major role for plasma lipids has been shown in melanoma risk [30,31], emerging data indicate that blood metabolites impact immune cell functions and clinical response to immunotherapy in melanoma patients [32]. The association of pre-treatment serum lipid levels and response to anti-PD-1 therapy was also reported in advanced intrahepatic cholangiocarcinoma [33]. A high body mass index, identifying overweight and obesity conditions, has been associated with a marked survival advantage after both targeted and immunotherapy in male melanoma patients [34], highlighting a potential link between factors associated with obesity, such as free FA, insulin/IGF1, and proinflammatory cytokines, and the activation of an effective anti-tumor immune response. Evidence has been provided that the anti-tumor activity of immune cells is facilitated by lipids, which are essential fuels and metabolic components [35]. In particular, FA metabolism processes are crucially involved in the survival and function of immune cells [36]. In this light, the higher levels of FA and SL we detected in LR compared to SR may define a favorable condition promoting targeted therapy-induced anti-tumor immune response.

Overall, our analysis of plasma lipid composition by targeted and untargeted approaches focusing on phospholipids reveals for the first time a different lipid pattern in melanoma patients and its potential predictive value for treatment response, suggesting that lipidomic profiles may uncover novel prognostic biomarkers for melanoma. However, this is a small explorative study suffering from some limitations. Specifically, the limitations of the study are the small number of patients and the lack of a validation cohort to support the obtained results. In addition, the retrospective nature of the present research requires the conduction of future prospective studies for further validation of its clinical applicability, for which the development of advanced and simplified techniques for lipid measurement is also necessary.

## 4. Materials and Methods

### 4.1. Plasma Samples

The study was approved by the Independent Ethic Committee and by the Institutional Review Board (protocol INT 94/18), and blood samples from cutaneous melanoma patients and control subjects were obtained upon written informed consent. The cohort of melanoma patients included groups at different clinical stages as reported in Table 1: (a) patients at early disease stages, at surgery for cutaneous primary melanoma (stage I-II, PT DH I/II, *n* = 36) or for regional lymph node metastases (stage III, PT III, *n* = 28) and (b) advanced stage metastatic patients (stage III–IV, PT III–IV) at the baseline of medical treatment with BRAF/MEK KI (*n* = 40) or with immune checkpoint inhibitors (ICI) (*n* = 40). Short-term (SR, ≤12 months) or long-term (LR, ≥13 months) responses to medical treatment in this class of patients were defined based on clinical controls during therapy. The series of control subjects included (a) patients at surgery for benign skin lesions (CTR DH, *n* = 75) and (b) blood donors from the hospital blood transfusion service (CTR HD, *n* = 40). Lipidomic analyses were carried out on a subgroup of samples from 20 patients with stage III–IV melanoma at baseline of KI therapy and 20 controls (CTR DH) matched for sex. Plasma samples were obtained from blood collected from fasting subjects in 10 mL vacutainer K_2_EDTA tubes [Becton Dickinson (Franklin Lakes, NJ, USA)] and centrifuged at 2000× *g* for 10 min at RT within 1h. To eliminate cell debris, plasma was centrifuged again at 2000× *g* for 10 min at RT before storage in aliquots at −80 °C until use. Samples were analyzed for levels of triglycerides (TG), total cholesterol (CHOL), and high-density lipoprotein (HDL) cholesterol by standard clinical laboratory methods; low-density lipoprotein (LDL) cholesterol concentrations were estimated as CHOL − (HDL + TG/5).

### 4.2. Measurement of FASN and DHCR24 Plasma Levels

Plasma levels of FASN and DHCR24 were quantified using ELISA as instructed by the manufacturer’s protocols (FASN, NBP2-82512, Novus Biologicals, Centennial, CO, USA; DHCR24, KTE62074, Abbkine, Wuhan, China). Plasma samples were tested undiluted. The limits of detection for FASN and DHCR24 were 0.31–20 ng/mL and 15–240 pg/mL, respectively. A microplate reader (Tecan Infinite M1000, Zürich, Switzerland) was adopted to determine the absorbance at 450 nm.

### 4.3. Chemicals and Reagents for Lipidomics

We purchased the following chemicals from Sigma-Aldrich (St. Louis, MO, USA): acetonitrile, 2-propanol, methanol, chloroform, formic acid, ammonium acetate, ammonium formate, and dibutylhydroxytoluene (BHT) were purchased. Purified water at a Milli-Q grade (Burlington, MA, USA) was used to prepare all aqueous solutions.

### 4.4. Untargeted Lipidomics

Lipids from plasma (25 µL) were analyzed by LC-MS/MS after dilution with water (75 µL) and extraction by a mixture of methanol/chloroform (850 µL, 2:1 *v*/*v*). LC-MS/MS consisted of a Shimadzu UPLC coupled with a Triple TOF 6600 Sciex (Concord, Toronto, ON, Canada) [37]. Analysis was in duplicate for all samples in positive and negative electrospray ionization. Acquisition of spectra was also carried out by full-mass scan from *m*/*z* 200–1500 and top-20 data-dependent acquisition from m/z 50–1500. A value of 50 eV was fixed for declustering potential, with a collision energy of 35 ± 15 eV. A reverse-phase Acquity CSH C18 column 1.7 μm, 2.1 × 100 mm (Waters, Franklin, MA, USA) was used for chromatographic separation by a gradient between (A) water/acetonitrile (60:40) and (B) 2-propanol/acetonitrile (90:10), both containing 10 mM ammonium acetate and 0.1% of formic acid.

### 4.5. Lipidomics Data Processing

The freeware software MS-DIAL (ver. 4.0) was used to attain the spectra deconvolution, peak alignment, blank filtering, and sample normalization. Values of 0.01 and 0.05 Da were set for MS and MS/MS tolerance for peak profile, respectively. By matching spectra with the LipidBlast database or in-house built mass spectral library identification was achieved. The Lawless algorithm was employed to normalize the intensities of analytes, excluding those with a CV% superior to 30% in the QC pool sample. The following lipid classes were considered: acylcarnitines (ACar), cholesterol (CHOL), cholesterol esters (CE), dihydroceramides (DHCer), ceramides (Cer), hexosylceramides (HexCer), lactosylceramides (LacCer), globotriaosylceramide (Gb3), sulfatides (SULF), gangliosides (GM3), sphingomyelins (SM), vitamin A fatty acid ester (VAE), N acyl glycil serine (NAGlySer), lysophosphatidylcholines (LPC), lysophosphatidylethanolamines (LPE), phosphatidylcholines (PC), phosphatidylethanolamine (PE), phosphatidylinositoles (PI), and plasmalogens, that are ether-linked phosphatidylcholines (EtherPC) and vinyl linked phosphatidylethanolamines (EtherPE). The total number of carbon atoms and degree of unsaturation (i.e., PC 40:2) or specific acyl chains detected (i.e., PC 18:0/18:1) are considered to name lipids.

### 4.6. Targeted Lipidomics for Sphingolipids and Free Fatty Acids Quantification

The extraction of sphingolipids (SL) was performed with the same method used for untargeted lipidomics, with the addition of an alkaline methanolysis step with the aim of ameliorating low abundant SL species recovery. An LC-MS/MS system consisting of a LC Dionex 3000 UltiMate (ThermoFisher Scientific, Waltham, MA, USA) coupled to a tandem mass spectrometer AB Sciex 3200 QTRAP (AB Sciex, Concord, Toronto, ON, Canada) equipped with electro spray ionization TurboIonSpray™ source operating in positive mode (ESI+) was employed to inject the methanol-based supernatant. Mobile phases (A) water + 0.2% formic acid + 2 mM ammonium formate and (B) methanol + 0.2% formic acid + 1 mM ammonium formate were used for chromatography [38]. An Acquity BEH C8 column 1.7 μm, 2.1 × 100 mm (Waters, MA, USA) was employed for the complex SL (DHCer, Cer, SM, HexCer, LacCer and GM3), whereas a Cortecs C18 1.6 μm, 2.1 × 100 mm (Waters, MA, USA) for sphingoid bases (Sph, S1P, dhSph and dhS1P), as previously detailed [38]. In addition, in the case of free FA, plasma (50 µL) deproteinization was carried out with 100 µL of cold isopropanol added with 10 µL of IS (2.5 µg/mL isobuthoxyacetic acid, 10 µg/mL undecanoic acid and 20 µg/mL heptadecanoic acid). The samples were centrifuged (12,000× *g*, 5 min), and supernatants (100 µL) were separated in glass vials for the derivatization of the carboxylic function. An aliquot of 50 µL of 50 mM of 3-NPH, 50 µL of 50 mM of EDC, and 50 µL of pyridine (7%) was added to the latter. The agents used for derivatization were prepared in a solution of 70% methanol. Derivatization was carried out in an incubator at 37 °C for up to 30 min. The solution was diluted with 250 µL 0.5% of formic acid in isopropanol and directly injected into the above-described apparatus. Mass spectrometry operated in negative electrospray ionization by multiple reaction monitoring mode and chromatographic separation was achieved on a Restek Raptor C18 2.7 µm 2.1 × 100 mm (Bellefonte, PA, USA) using mobile phase (A) water + 0.1% formic acid and (B) acetonitrile. The following elution program (%B) was used: 0–2.5 min 10%, 2.5–31 min 10–99%, 31–35 99%, 35–35.2 99–10% maintained until 42 min; with the flow rate of 0.4 mL/min; the column and the autosampler temperature were 35 °C and 15 °C.

### 4.7. Statistical Analysis

The main characteristics of the study cohort were described by standard descriptive statistics, medians and ranges for continuous variables, and frequency tables for categorical variables. For continuous variables, the one-way analysis of variance was adopted to assess the relationship between the continuous variables and the different study outcomes. To adjust for group comparisons the Turkey correction was adopted. To assess the association between FASN and DHCR24 plasma levels with the disease status (cases vs. controls), logistic regression models [39] were also performed, and results were reported in terms of odds ratio (OR) and 95% confidence intervals (95%CI). The Pearson correlation coefficient (r) and its 95%CI were used to assess the strength of the association of the considered biochemical variables [40]. For untargeted lipidomic analyses, data were checked for integrity, filtered by interquartile range, and log-transformed before the statistical analysis. The Wilcoxon exact test (W) [41], followed by false discovery rate (FDR) adjustment, was adopted to highlight differentially expressed lipids in melanoma patients and controls as well as between treatment responses (LR vs. SR). The same approach was adopted for the analysis at the lipid family level. For the latter, the principal component approach (PCA) [42] was used to combine lipids of the same family. For targeted lipidomic analysis, plasma concentrations expressed as µM were considered and analyzed according to the W-test. For SL family analysis, the sum of the lipids with FA between C16 and C24 was considered, whereas that of saturated and unsaturated free FA between C16 and C18 was considered for the sphingoid bases and their phosphate form. A w-test followed by FDR adjustment was applied to determine families differentially expressed between melanoma patients and controls as well as between treatment responses. SAS software (Version 9.4, SAS Institute Inc., Cary, NC, USA), adopting a nominal level of 5%, was used for all statistical analysis, and R-software (R version 4.2.1, Foundation for Statistical Computing, Vienna, Austria) with the ggplot2 package (version 3.4.2) for graphical representations.

## 5. Conclusions

The identification of changes in lipid profiles, providing valuable biomarkers, and shedding light on mechanisms of diseases could have a crucial role in patients’ risk stratification and precision medicine, allowing for the selection and customization of medical treatment based on individual lipid profiles. In this pilot study, we evaluated whether a specific lipid pattern characterizes plasma from melanoma patients and their response to targeted therapy with KI. Taken together, our results showed that altered levels of selected lipid species are detectable in melanoma patients’ plasma specifically in patients with a more favorable prognosis. Although these results were obtained in a small patient cohort, they paved the way for the hypothesis that lipidomic profiles may uncover novel prognostic biomarkers for melanoma.

## Figures and Tables

**Figure 1 ijms-25-01558-f001:**
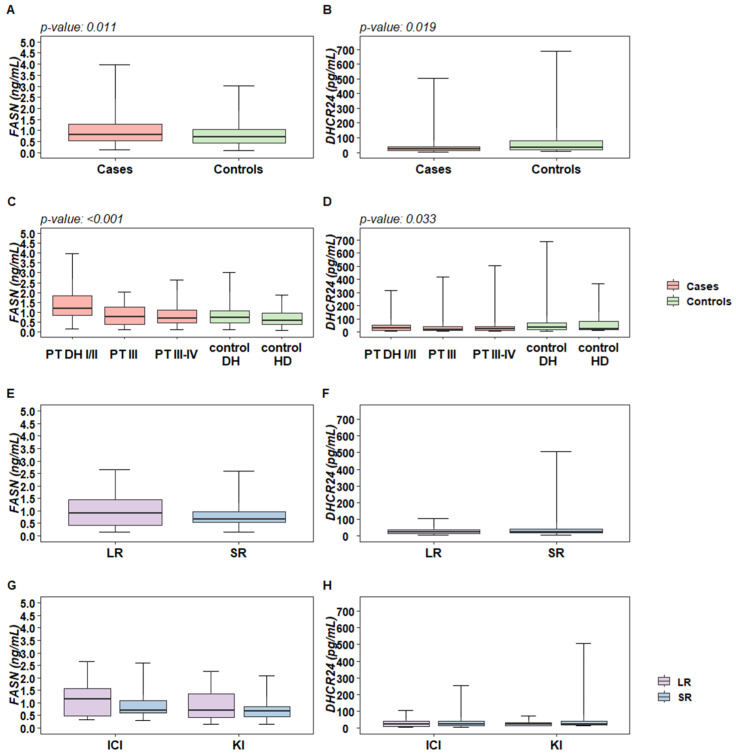
Distribution of FASN and DHCR24 levels in melanoma patients and control subjects. (**A**,**B**) Box plots reflecting the distribution of FASN (ng/mL) and DHCR24 (pg/mL) in melanoma patients and control subjects in (**C**,**D**) when stratified in groups according to clinical stages and as DH (day-hospital) or HD (healthy donors) for controls groups. (**E**,**F**) Box plots reflecting the overall distribution of FASN and DHCR24 in melanoma patients according to short-term (SR, ≤12 months) or long-term (LR, ≥13 months) response to treatment, and (**G**,**H**) stratified by medical treatment (ICI—immune checkpoint inhibitors; KI—kinase inhibitors). Each box indicates the 25th and 75th centiles. The horizontal line inside the box indicates the median and the whiskers indicate the extreme measured values. The reported *p*-values refer to ANOVA significant results.

**Figure 2 ijms-25-01558-f002:**
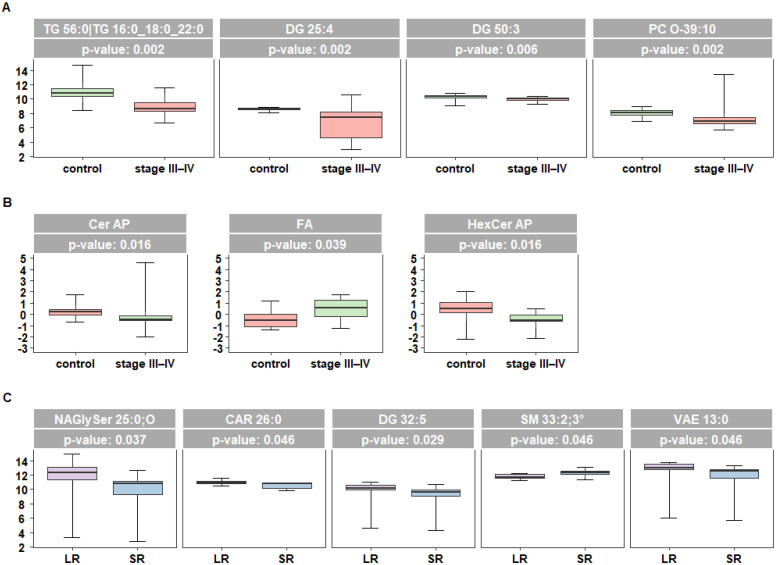
Untargeted lipidomic analysis of plasma samples from 20 melanoma patients at advanced stages (stage III–IV) and 20 sex-matched control subjects (control). Box plots reflecting the distribution of the differentially expressed lipids between cases and controls (**A**) at the single lipid level and (**B**) at the lipid family level. (**C**) Box plots reflecting the distribution of the differentially expressed lipids between LR and SR patients (short-term SR, ≤12 months, or long-term LR, ≥13 months). Each box indicates the 25th and 75th centiles. The horizontal line inside the box indicates the median and the whiskers indicate the extreme measured values expressed as µM. The reported *p*-values refer to FDR-adjusted Wilcoxon significant results.

**Figure 3 ijms-25-01558-f003:**
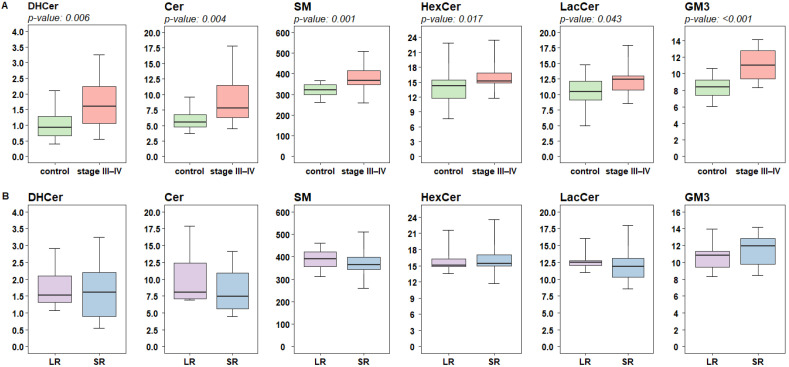
Targeted lipidomic analysis of complex sphingolipids levels in plasma samples from 20 melanoma patients at advanced stages (stage III–IV) and 20 sex-matched control subjects (control). Box plots reflecting the distribution of plasma sphingolipid profiles at lipid family level (**A**) in cases and controls, and (**B**) in patients according to treatment response (short-term SR, ≤12 months, or long-term LR, ≥13 months). Each box indicates the 25th and 75th centiles. The horizontal line inside the box indicates the median and the whiskers indicate the extreme measured values, expressed as µM. The reported *p*-values refer to FDR-adjusted Wilcoxon significant results.

**Figure 4 ijms-25-01558-f004:**
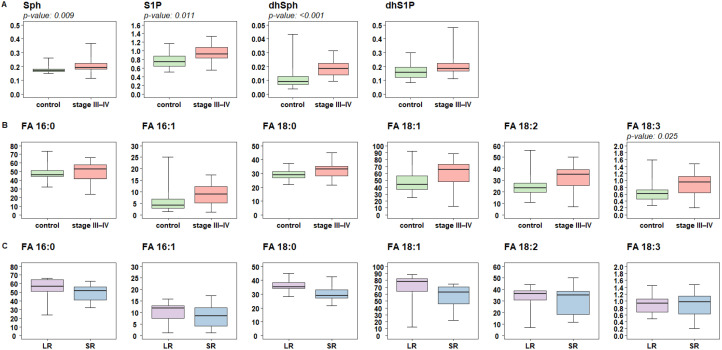
Targeted lipidomic analysis of sphingoid bases and their phosphate form and of saturated and unsaturated free fatty acids between C16 and C18 in plasma samples from 20 melanoma patients at advanced stages (stage III–IV) and 20 sex-matched control subjects (control). Box plots showing (**A**) the distribution of plasma sphingoid-bases profiles in cases and controls, (**B**) of levels of free FA in plasma of cases and controls, and (**C**) in patients according to treatment response (short-term SR, ≤12 months, or long-term LR, ≥13 months). Each box indicates the 25th and 75th centiles. The horizontal line inside the box indicates the median and the whiskers indicate the extreme measured values, expressed as µM. The reported *p*-values refer to FDR-adjusted Wilcoxon significant results.

**Table 1 ijms-25-01558-t001:** Study sample characteristics.

	Frequency	Percent
Sex		
F	107	41.31
M	152	58.69
Patient’s Group		
PT DH I/II	36	13.9
PT III	28	10.81
PT III–IV	80	30.89
control DH	75	28.96
control HD	40	15.44
Treatment type *(PT III–IV only)*		
ICI *(IPI/NIVO)*	40	50
KI *(BRAF/MEK)*	40	50
Response class *(PT III–IV only)*		
SR	55	68.75
LR	25	31.25
Age *n; median (range)*	256; 50 (17–85)	
Triglycerides (mg/dL) *n; median (range)*	258; 114 (40–528)	
Total cholesterol (mg/dL) *n; median (range)*	258; 194 (69–336)	
LDL (mg/dL) *n; median (range)*	258; 111.6 (5.8–242.8)	
HDL (mg/dL) *n; median (range)*	258; 53 (12–227)	
DHCR24 (pg/mL) *n; median (range)*	259; 24.78 (3.61–687.82)	
FASN (ng/mL) *n; median (range)*	258; 0.74 (0.08–3.97)	

DH—day-hospital; HD—healthy donors; ICI—immune checkpoint inhibitors; KI—kinase inhibitors; SR—short-term responders (≤12 months); LR—long-term responders (≥13 months).

**Table 2 ijms-25-01558-t002:** Plasma levels of FASN, DHCR24, TG, CHOL, LDL, and HDL: results from univariate logistic regression models according to disease status.

	OR (95% CI) *	OR (95% CI) **
DHCR24 (pg/mL)	0.74 (0.56; 0.96)	0.84 (0.62; 1.13)
FASN (ng/mL)	1.43 (1.09; 1.88)	1.24 (0.92;1.67)
TG (mg/dL)	1.26 (0.97; 1.64)	1.04 (0.79; 1.37)
Total CHOL (mg/dL)	0.97 (0.76; 1.24)	0.77 (0.58; 1.03)
LDL (mg/dL)	0.99 (0.78; 1.27)	0.83 (0.62; 1.10)
HDL (mg/dL)	0.77 (0.58; 1.03)	0.78 (0.56; 1.07)

* Odds ratio computed for a specific unit change for each variable equal to one SD; ** Odds ratio computed for a specific unit change for each variable equal to one SD and adjusted by age.

## Data Availability

The original contributions presented in the study are included in the article; further inquiries can be directed to the corresponding author.

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
