# Peer review of "Alterations in Plasma Lipid Profiles Associated with Melanoma and Therapy Resistance"

_ijms, 2024, doi:10.3390/ijms25031558_

Round 1

Reviewer 1 Report

Comments and Suggestions for Authors

The manuscript entitled “Alterations in plasma lipid profiles associate with melanoma 2 and therapy resistance”  by Cas et al  was reviewed initially. Figures are not visible  Its hard to see the results. The authors should supply visible figures with proper statistics so that it can be reviewed.

Minor English correction is required and should be checked throughout the manuscript. For eg.

“Plasma samples from patients and controls …. controls (n=115)”

“By capturing the intricate and …. of an organism”

Comments on the Quality of English Language

NO

Author Response

1) Figures are not visible.  It is hard to see the results. The authors should supply visible figures with proper statistics so that it can be reviewed.

Reply. We agree with the reviewer. Figures have been implemented and statisctics introduced as requested.

2) Minor English correction is required and should be checked throughout the manuscript. For eg.

“Plasma samples from patients and controls …. controls (n=115)”

“By capturing the intricate and …. of an organism”

Minor editing of English language required

Reply. We apologize for the mistakes. Manuscript has been edited for language and typos. 

Reviewer 2 Report

Comments and Suggestions for Authors

Very interesting and well written article 

I have some considerations for the authors that are necessary in my opinion for publication

1) Why does the materials and methods section come after the discussion? This way the reading becomes confusing and inaccurate, it needs to be reversed

2) add paragraph limitations of the study

3) deepen the introduction better by talking about melanoma and new therapies available to date, I leave some refs for useful authors,

- Villani, A., Scalvenzi, M., Micali, G., Lacarrubba, F., Fornaro, L., Martora, F., & Potestio, L. (2023). Management of Advanced Invasive Melanoma: New Strategies. Advances in therapy, 40(8), 3381-3394. https://doi.org/10.1007/s12325-023-02555-5.

Villani, A., Potestio, L., Fabbrocini, G., Troncone, G., Malapelle, U., & Scalvenzi, M. (2022). The Treatment of Advanced Melanoma: Therapeutic Update. International journal of molecular sciences, 23(12), 6388. https://doi.org/10.3390/ijms23126388.

4) Minor editing of English language required.

5) various typos in the text, need to proofread all of it 

6) Better check the data entered in the tables 

Comments on the Quality of English Language

Minor editing of English language required

Author Response

Very interesting and well written article

Reply: We thanks the reviewer for the positive comment.

I have some considerations for the authors that are necessary in my opinion for publication

1) Why does the materials and methods section come after the discussion? This way the reading becomes confusing and inaccurate, it needs to be reversed

Reply. We agree with the reviewer. The paragraph Conclusions has been moved below the Discuussion section.

2) add paragraph limitations of the study

Reply. A paragraph summarizing the limitations of the study has been introduced as requested.

3) deepen the introduction better by talking about melanoma and new therapies available to date, I leave some refs for useful authors,

Villani, A., Scalvenzi, M., Micali, G., Lacarrubba, F., Fornaro, L., Martora, F., & Potestio, L. (2023). Management of Advanced Invasive Melanoma: New Strategies. Advances in therapy, 40(8), 3381-3394. https://doi.org/10.1007/s12325-023-02555-5.

Villani, A., Potestio, L., Fabbrocini, G., Troncone, G., Malapelle, U., & Scalvenzi, M. (2022). The Treatment of Advanced Melanoma: Therapeutic Update. International journal of molecular sciences, 23(12), 6388. https://doi.org/10.3390/ijms23126388.

Reply. We thanks the reviewer for the suggestion. The Introduction has been implemented adding a brief summary of the new therapies available for the clinical management of melanoma patients. References have been updated accordingly (Ref. 1 and 2).

4) Minor editing of English language required.

Reply. We apologize for the errors. Manuscript has been edited for language. 

5) Various typos in the text, need to proofread all of it

Reply. Typos have been corrected.

6) Better check the data entered in the tables

Reply. Tables have been revised as requested. 

Round 2

Reviewer 2 Report

Comments and Suggestions for Authors

Article is improved after revisions

In my opinion is suitable for publication